# Comparison of the Trilateral Flash Cycle and Rankine Cycle with Organic Fluid Using the Pinch Point Temperature

**DOI:** 10.3390/e21121197

**Published:** 2019-12-05

**Authors:** Kai-Yuan Lai, Yu-Tang Lee, Miao-Ru Chen, Yao-Hsien Liu

**Affiliations:** 1Department of Mechanical Engineering, National Chiao Tung University, 1001 Ta-Hsueh Road, Hsinchu 30010, Taiwan; itria20076@itri.org.tw (K.-Y.L.); leeadam0710@gmail.com (Y.-T.L.); 2Green Energy and Environment Research Laboratories, Industrial Technology Research Institute (ITRI), Sec. 4, Zhongxing Road, Hsinchu 310, Taiwan; MiaoRuChen@itri.org.tw

**Keywords:** organic Rankine cycle, trilateral flash cycle, pinch point

## Abstract

Low-temperature heat utilization can be applied to waste heat from industrial processes or renewable energy sources such as geothermal and ocean energy. The most common low-temperature waste-heat recovery technology is the organic Rankine cycle (ORC). However, the phase change of ORC working fluid for the heat extraction process causes a pinch-point problem, and the heat recovery cannot be efficiently used. To improve heat extraction and power generation, this study explored the cycle characteristics of the trilateral flash cycle (TFC) in a low-temperature heat source. A pinch-point-based methodology was developed for studying the optimal design point and operating conditions and for optimizing working fluid evaporation temperature and mass flow rate. According to the simulation results, the TFC system can recover more waste heat than ORC under the same operating conditions. The net power output of the TFC was approximately 30% higher than ORC but at a cost of higher pump power consumption. Additionally, the TFC was superior to ORC with an extremely low-temperature heat source (<80 °C), and the ideal efficiency was approximately 3% at the highest work output condition. The TFC system is economically beneficial for waste-heat recovery for low-temperature heat sources.

## 1. Introduction

In the past, low-temperature heat recovery was not widely used due to low energy prices and technical barriers. Now, because of an awareness of the energy crisis and an interest in sustainability, the development of clean energy and energy-saving technologies are priority research topics in every country. Waste heat is usually a 24-hour discharged heat from industrial processes, and approximately 63% of the waste heat is <100 °C [1]. Efficient waste-heat recycling improves economic efficiency, energy conservation, and waste-heat reduction. Among all the waste-heat utilization technologies, the organic Rankine cycle (ORC) is a mature technology that features high reliability and low cost. It is arguably the most efficient energy conversion solution for low-temperature waste-heat power generation at present [2,3], and it is widely used to recover industrial waste heat [4,5,6,7], geothermal energy [8,9], biomass heat, and solar thermal energy [10,11,12]. However, the heat extraction efficiency of the ORC remains low because the working fluid cannot obtain a higher temperature than that of the heat source from the phase change [13]. The pinch point limits the energy capture efficiency and increases waste heat [14].

To achieve efficient energy extraction, Smith proposed the trilateral flash cycle (TFC) in 1993 [15]. The non-isothermal and phase-unchanged endothermic process in a TFC cycle solves the pinch-point problem of the ORC. Although the thermal efficiency of the TFC is slightly lower than that of the ORC [16], the exergy efficiency is relatively higher. The high exergy efficiency is more influential than the high thermal efficiency under low-temperature heat source conditions [17]. The potential heat recovery rate of the TFC is 14–85% higher than that of the ORC [17,18]. The TFC exhibits more favorable power generation capacity under low-temperature heat source conditions (<80 °C) [19].

The TFC system is an alternative solution to low-temperature sites for energy saving and economic benefits. It is aimed at promoting the utilization of waste heat in future power generation. In order to maximize the benefits of the system, system optimization in terms of efficiency is worth exploring. Numerous studies were conducted to select the favorable working fluids for a designated heat source, which is one of the most important phases in building TFC and ORC systems [20,21]. 

The location of the heat transfer pinch point in the evaporator is also critical for determining the operating parameters [22,23,24]. However, different inlet and outlet temperatures of heat source, together with diverse working fluids, would lead to a variety of combinations. Therefore, the establishment of matching rules can determine the appropriate working fluids and operating conditions for a given heat source. In the present study, a physical mathematical model seeking the location of the pinch point is developed. The operating parameter selections for TFC and ORC are also discussed. 

Determination of the location of the pinch point has an important impact on the optimum mass flow rate and pressure of the TFC system. This study investigated the influence of pinch points from the heat extraction process of TFC and ORC systems for low-temperature waste-heat sources. The pinch-point temperature between the heat source and the working fluid was 5 °C [25,26]. The study investigated the optimal working fluid and operating conditions for TFC and ORC systems by evaluating the evaporation temperature and mass flow rate using the pinch-point temperature. The heat extraction rate and net power output between the two cycles were also compared.

Based on our experience in building ORC systems (Figure 1), the test results showed that the net power is 193 kW and the thermal efficiency is only 4.701% under the condition that the heat source temperature is 82 °C/189 TPH(Ton per hour) [27]. The TFC system developed in this study has the potential to improve power generation.

## 2. Theoretical Modeling

The thermodynamics cycle models of the ORC and TFC were established according to the selected components, and the properties were evaluated using the software REFPROP 9. Figure 2a presents a traditional ORC T–s diagram. The working fluid was pumped to a high-pressure state (point 1 to 2) and was heated to the superheated state after passing through a heat exchanger in a phase change process (point 2 to 3). The high-pressure, superheated fluid then underwent gaseous expansion without a phase change (point 3 to 4), and the work was produced during this expansion. Afterward, the fluid was condensed back to the original low-pressure liquid state to complete the cycle. Figure 2b illustrates a typical TFC thermodynamic cycle. The differences between the two systems are the heat exchange (point 2 to 3) and expansion (point 3 to 4) processes. For the TFC, the working fluid was pumped and heated to become a high-pressure, saturated liquid without phase change (point 3). The high-pressure, saturated liquid was expanded to produce work, and the wet or dry vapor state was attained at the expander exit (point 4). 

In the two systems, the fixed parameters are the temperature and the mass flow rate of heat source, which are 80 °C and 4.16 kg/s, respectively. The heat sink temperature is maintained at 30 °C. The pump power consumption can be calculated using the pressure difference and specific volume (v) of the working fluid (${ŋ}_{p}$ is the isentropic efficiency of the pump). The operating temperature at the inlet of the pump (point 1) was 35 °C, which was evaluated according to the condenser inlet temperature (point 7) of 30 °C.
(1)$${\dot{W}}_{p}=\frac{{\dot{m}}_{f}v\left(P_{2}-P_{1}\right)}{{ŋ}_{p}}.$$

The heat exchanger was assumed to be adiabatic with no frictional loss. For the ORC, the property at the exchanger exit (point 3) was evaluated using the pressure and pinch temperature of the superheated gas. For the TFC, the property at the exchanger exit (point 3) was obtained using the saturated state. In the TFC heating process (2–3), the temperature of the heat source was settled (state 5, 80 °C), and the pinch point was determined according to the saturation properties of state 3. During the simulation, the pinch-point temperature difference was determined using the minimum temperature difference observed during the heat exchange process. The heat transfer rate was calculated using the enthalpy change during the heating process.
(2)$${\dot{Q}}_{H}={\dot{m}}_{f}\left(h_{3}-h_{2,a}\right)={\dot{m}}_{hw}(h_{6}-h_{5}).$$

During the two-phase expansion process in TFC, the work output was evaluated using the enthalpy change and isentropic efficiency of the nozzle (${ŋ}_{nozzle}$) and the rotor (${ŋ}_{rotor}$).
(3)$${\dot{W}}_{nozzle}={\dot{m}}_{f}\left(h_{3}-h_{4,s}\right){ŋ}_{nozzle}={\dot{m}}_{f}\left(h_{3}-h_{4,a}\right).$$
(4)$${\dot{W}}_{expander,TFC}={\dot{W}}_{nozzle}\cdot {ŋ}_{rotor}.$$

The equations of the two-phase expansion are based on Reference [28].
(5)$${ŋ}_{nozzle}=0.865+0.00175\cdot d_{v},$$
(6)$${ŋ}_{rotor}=0.575+0.325\cdot x_{4},$$
where $d_{v}$ is the saturated vapor density at low pressure of condensation, and $x_{4}$ is the outlet dryness of the expander.

The expansion process in ORC features a single phase. The electric power produced in the expander can be calculated by the expander efficiency (${ŋ}_{expander}$).
(7)$${\dot{W}}_{expander,ORC}={\dot{m}}_{f}\left(h_{3}-h_{4,s}\right){ŋ}_{expander}={\dot{m}}_{f}\left(h_{3}-h_{4,a}\right).$$

The condenser was assumed to be adiabatic with no frictional loss.
(8)$${\dot{Q}}_{L}={\dot{m}}_{f}(h_{4,a}-h_{1})={\dot{m}}_{cw}(h_{8}-h_{7}).$$

The thermal efficiency of the system can be estimated using net power and heat transfer.
(9)$${\dot{W}}_{net}={\dot{W}}_{expander}-{\dot{W}}_{p}.$$
(10)$${ŋ}_{th}=\frac{{\dot{W}}_{net}}{{\dot{Q}}_{H}}.$$

The exergy efficiency can also be determined as follows: (11)$${\dot{Q}}_{max}\;=\;{\dot{m}}_{hw}C_{pw}\Delta T_{sysrtem},$$
(12)$$\Delta T_{system}=T_{max}-T_{min},$$
(13)$${ŋ}_{E}=\frac{{\dot{Q}}_{H}}{{\dot{Q}}_{max}},$$
where $C_{pw}$ is the specific heat of hot water, $T_{max}$ is the heat source temperature, and $T_{min}$ is the heat sink temperature.

To optimize this thermal model, the mass flow rate of the working fluid (${\dot{m}}_{f}$) was adjusted according to the pinch point of the heat exchange process. By using the predetermined heat source inlet temperature, the saturated evaporation temperature was obtained. Figure 3 shows the heat transfer process with temperature in the ORC and TFC. Reducing the mass flow rates causes a relatively high heat source temperature at the outlet, indicating that the energy of the heat source is not used efficiently. This leads to a higher temperature at the heat exchanger outlet $\left(1-2^{\prime }\right)$. Increasing the mass flow rate can cause more heat absorption and increase power generation; however, it reduces the pinch-point temperature difference, leading to a lower temperature at the heat exchanger outlet $\left(1-2^{\prime \prime }\right)$. The maximal mass flow was determined such that the pinch-point temperature difference was at least 5 °C; the temperature at the heat exchanger outlet was 2 °C, and this temperature difference was maintained to ensure the effective heat exchange process. Table 1 lists the test parameters and working fluids of the ORC and TFC systems. Hot water at 80 °C was used as the heat source, and the mass flow rate was 15 tons/h. Cold water at 30 °C was used as the heat sink. The commonly used working fluid in the low-temperature power generation system was selected according to the literature, including R245fa, R134a, R236fa, and R1233zd. Figure 4 shows the flowchart of the simulation. The properties and mass flow rate (${\dot{m}}_{f}$) of the working fluid could be calculated using the aforementioned process. 

## 3. Results

Figure 5 shows the simulation results of the TFC system. The results indicated that the net power and thermal efficiency of the TFC increased with the evaporation temperature. According to the net power and efficiency results, R245fa and R1233zd had comparable performance, and its higher operating pressures and pump consumption limitations caused the low performance of R134a. The effect of mass flow rate and evaporation temperature on net power in the TFC system is shown in Figure 6. The results indicated that the mass flow rate of the working fluid decreased with the increasing evaporation temperature because of the pinch temperature limitation between heat exchangers. The heat transfer rate of the system increased with the evaporation temperature, and it reduced the temperature of the heat source outlet. To obtain the desirable temperature difference of the heat exchanger, the mass flow rate and pinch-point temperature were controlled accordingly. The optimal mass flow rate of each working fluid at various evaporation temperatures was identified to calculate the maximum net power of the system. The results indicated that the required mass flow rate for R245fa was smaller than that for R1233zd, because R245fa produces a high enthalpy difference per unit mass at the same saturation temperature, leading to a relatively low operation cost.

In the TFC system, the net power and the efficiency of the working fluid as a function of the mass flow rate are shown in Figure 7. The results indicated that the net power and efficiency initially increased with the mass flow rate at the fixed heat source temperature. However, the net power and efficiency eventually reached a maximal value and then decreased with the increasing mass flow rate. Because the increasing mass flow rate enlarged the heat transfer, the heat source outlet temperature was reduced to achieve the minimum pinch-point temperature difference. Therefore, the evaporation temperature was adjusted to respond; a decrease in the evaporation temperature resulted in a reduction in the enthalpy of the working fluid in the system, which affected the net power and thermal efficiency. For the minimum allowable pinch-point temperature difference, it was observed that the increasing working fluid mass flow rate only increased the total net power output. When the mass flow rate exceeded the desirable range, reducing the system evaporation temperature caused a decrease in performance. The maximal desirable mass flow rate of R134a was considerably smaller than that of the other working fluids because it produced the largest amount of heat exchange. 

The optimal operating conditions and simulation results of each working fluid in the TFC system are presented in Table 2. The results indicated that the evaporation temperature was set to a maximum value of 75 °C (pinch temperature and heat source were 5 °C and 80 °C, respectively). The net power and efficiency of each working fluid could reach the optimum operating conditions, and the TFC system exhibited favorable heat extraction efficiency. However, the phase change in the ORC system caused the pinch-point problem during the heat exchange process, which limited the endothermic process of the system. Table 2 also presents the high gross power output of R134a; however, its high working pressure limited its application. The high pump consumption caused an unfavorable net power output compared with other working fluids. Consequently, R245fa exhibited the highest overall net power and efficiency, followed by R1233zd.

The simulation results of the ORC system are shown in Figure 8. The system efficiency increased with increasing evaporation temperature, but the optimal net power was observed at a moderate evaporation temperature. When the evaporation temperature was higher than the optimal point, the net power of the system decreased because the evaporation temperature of the ORC system was limited by the pinch point with a fixed heat source outlet. To maintain the pinch point temperature difference, the excessive evaporation temperature reduced the working fluid mass flow rate, leading to the low net power. The results indicated that, under this condition, R1233zd was superior to other working fluids in terms of net power generation or net efficiency. Figure 9 shows that a decrease in the mass flow rate of each working fluid gradually reduced the net power. Additionally, R1233zd generated the largest net power with the lowest mass flow rate compared to the other working fluids.

The optimal operating conditions and simulation results of each working fluid in the ORC system are presented in Table 3. The net power and efficiency of R1233zd were the highest, and the evaporation temperature of the system under the optimal operating conditions was 55–56 °C. The pump consumption of R134a in the ORC system was lower than that of the TFC system; however, its net output was inferior to R245fa and R1233zd. Overall, the effect of working fluid on the net efficiency in the ORC system was less valuable than that in the TFC system, and its net power performance was lower than that of the TFC system. Notably, when the ORC was in the optimal operating conditions, the required flow rate of the working fluid for the ORC was much smaller than that for the TFC system, yielding less construction costs and a small volume expansion unit.

## 4. Discussion

The optimal simulation results for each working fluid of the TFC and ORC systems are compared in Figure 10. Under the same conditions, the power generation of all the working fluids in the TFC was twice that of the ORC. The TFC system required higher evaporation pressure for a favorable operating condition, which also caused substantially higher pump consumption in the TFC system than that in the ORC system. One of the focuses of future studies on the TFC will be a methodology to effectively reduce pump consumption.

Figure 11 shows the net power and heat transfer for the TFC and ORC systems at different pinch temperatures. To compare the difference between the two systems, R245fa and R1233zd were selected under the most favorable performance conditions of the TFC and ORC. According to the simulation results, the net power and heat transfer were negatively correlated with the pinch-point temperature. When the pinch-point temperature difference decreased, the system absorbed more heat in the cycle and also generated a larger amount of power. The TFC system exhibited higher heat transfer and net power than the ORC system at various pinch-point temperatures, which indicated that the TFC can be applied for large-scale power generation.

The most favorable simulation results for the TFC and ORC systems are shown in Figure 12. Figure 12a shows that the maximum net power generated by the TFC was 18.74 kW, which, with a high pump consumption, was much higher than the ORC’s power generation of 12.51 kW. Figure 12b shows that the TFC system exhibited a high energy usage rate because of the relatively small pinch temperature limitation. According to the cold and heat source conditions of the environment, the estimated exergy in the system was 871 kW, and the heat transfer of the TFC and ORC systems was 692 kW and 330 kW, respectively. This indicated that the optimal exergy efficiency of the TFC and ORC systems was 79.4% and 37.8%, respectively.

## 5. Conclusions

The study investigated the pinch-point temperature problem by manipulating the evaporation temperature and mass flow rate of various working fluids and evaluated the performance of the TFC and ORC systems for a low-grade heat source; the findings are as follows:
(1)The evaporation temperature influenced the net power and thermal efficiency of the TFC system. When the temperature increases, each working fluid can achieve relatively large power generation and thermal efficiency. The pinch temperature relatively limits the operating maximum temperature of the ORC system. The excessive evaporation temperature reduces the working fluid mass flow rate and system performance.(2)Under fixed environmental conditions, the increase in mass flow rate has a limit, and the mass flow rate decreases with the evaporation temperature. When the mass flow rate is higher than the optimum value, the net power generation of the system decreases. Additionally, increasing the working fluid mass flow rate only increases the net power at the minimum permissible pinch-point temperature difference and does not increase thermal efficiency. Thermal efficiency only changes with the evaporating temperature adjustment. At optimal operating conditions, the ORC system requires a lower working fluid mass flow rate than the TFC system.(3)Net power and heat transfer are negatively correlated with the pinch temperature. When the pinch point is close to the minimum temperature, more heat transfer is intercepted from the system, which can generate larger net power. The heat exchange efficiency of the ORC system is limited by the pinch point generated from the phase change of the working fluid. By contrast, the efficiency of the TFC system is relatively high because the TFC is non-isothermal and phase-unchanged, which gives it a more favorable heat extraction rate for the heat source.(4)From the simulation results, under the same heat source condition of 80 °C, the thermal efficiency of each working fluid in the ORC was compared with that in the TFC, and the TFC system exhibited a more favorable heat extraction rate and higher pump consumption, and the net power output was approximately 30% higher than the ORC system.

## Figures and Tables

**Figure 1 entropy-21-01197-f001:**
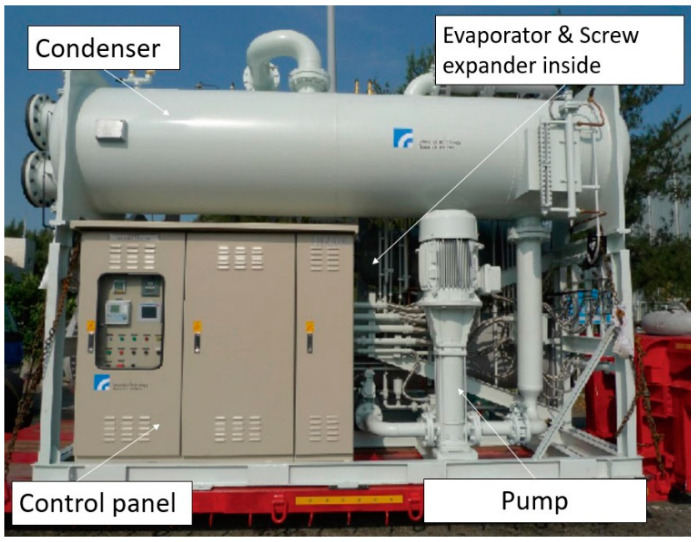
A photograph of the 200-kW organic Rankine cycle system. Source: Industrial Technology Research Institute (ITRI).

**Figure 2 entropy-21-01197-f002:**
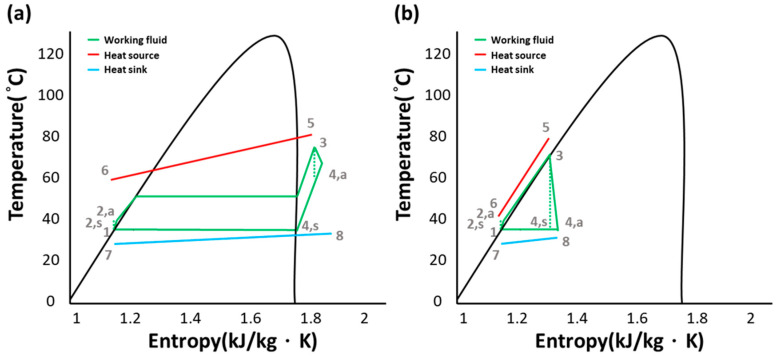
(**a**) T–s diagram for organic Rankine cycle (ORC); (**b**) T–s diagram for trilateral flash cycle (TFC).

**Figure 3 entropy-21-01197-f003:**
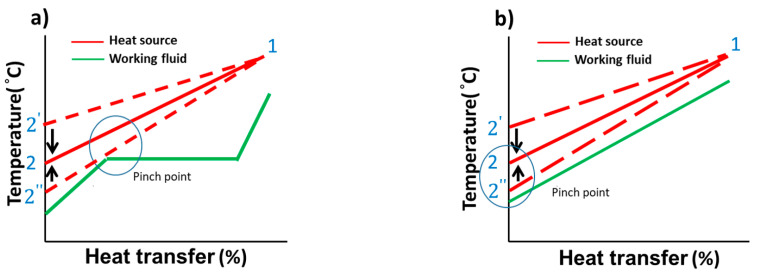
(**a**) Temperature–heat transfer in the ORC; (**b**) temperature–heat transfer in the TFC.

**Figure 4 entropy-21-01197-f004:**
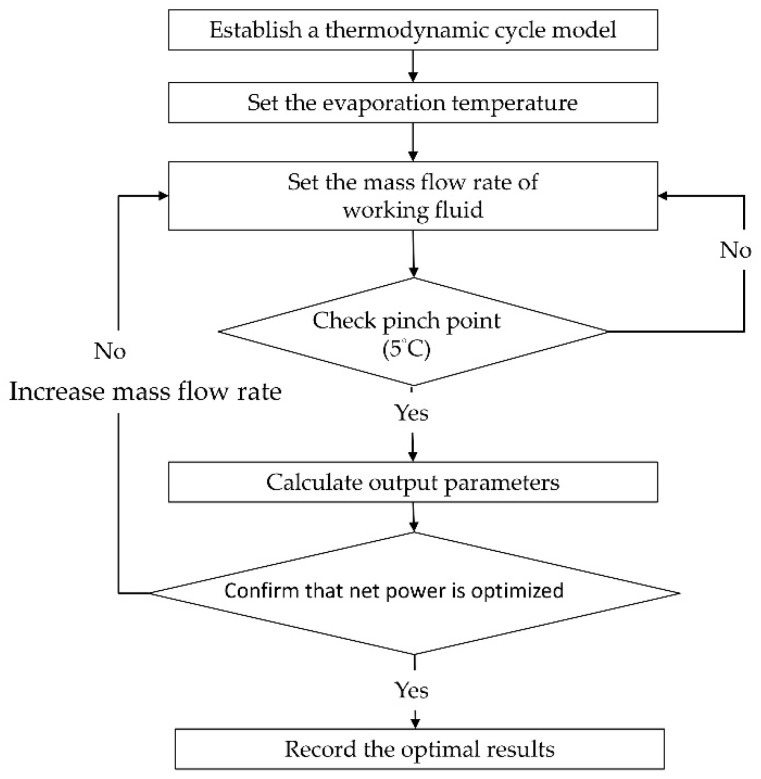
Flowchart of the simulation procedure.

**Figure 5 entropy-21-01197-f005:**
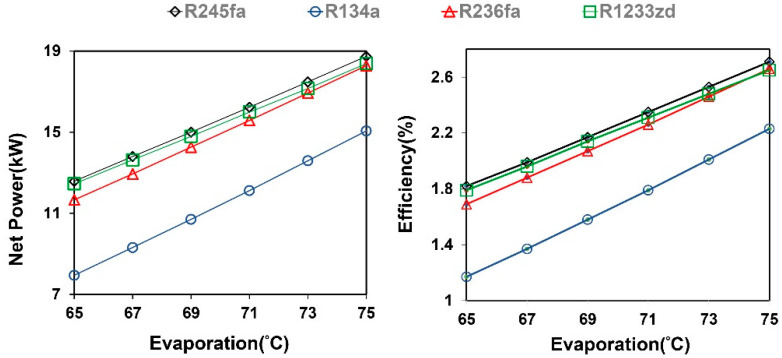
Effect of evaporation temperature on net power generation and efficiency in the TFC system.

**Figure 6 entropy-21-01197-f006:**
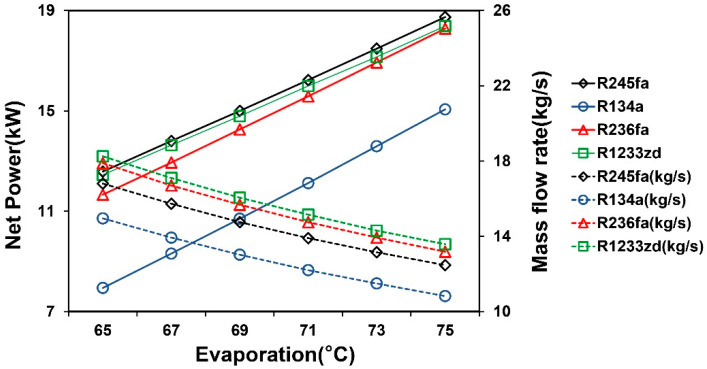
Finding the most favorable power generation by fine-tuning the mass flow rate and evaporation temperature in the TFC system.

**Figure 7 entropy-21-01197-f007:**
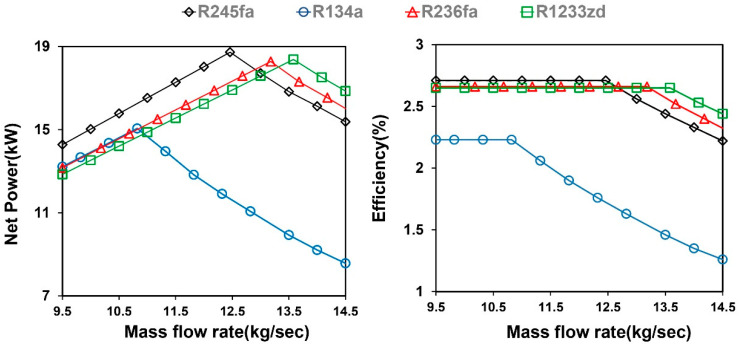
Net power and efficiency of the TFC system for different mass flow rates (80 °C hot water).

**Figure 8 entropy-21-01197-f008:**
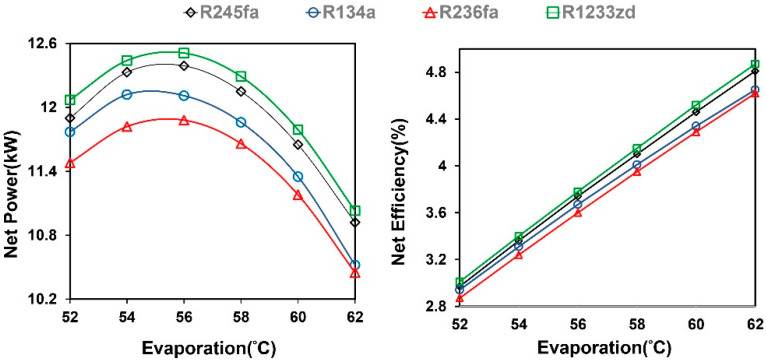
Effect of evaporation temperature on net power generation and efficiency in the ORC system.

**Figure 9 entropy-21-01197-f009:**
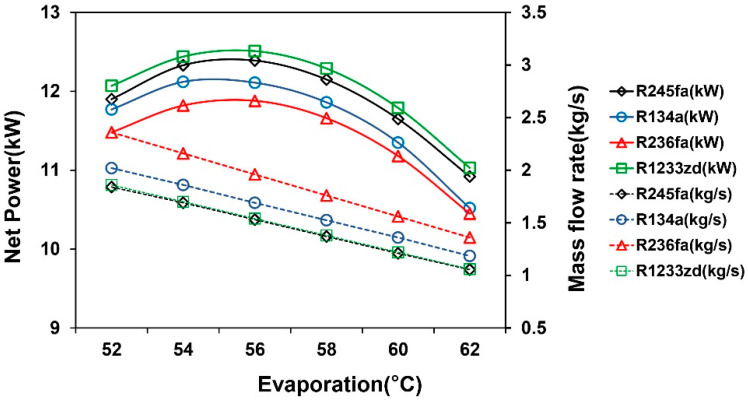
Finding the most favorable power generation by fine-tuning the mass rate and evaporation temperature in the ORC system.

**Figure 10 entropy-21-01197-f010:**
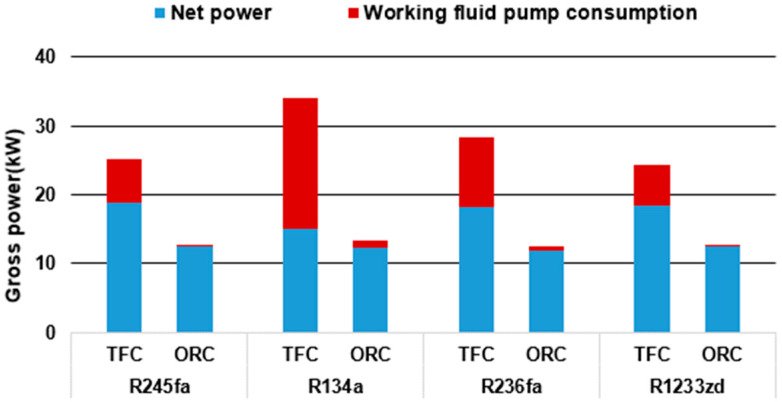
Gross power for the TFC and ORC with different working fluids.

**Figure 11 entropy-21-01197-f011:**
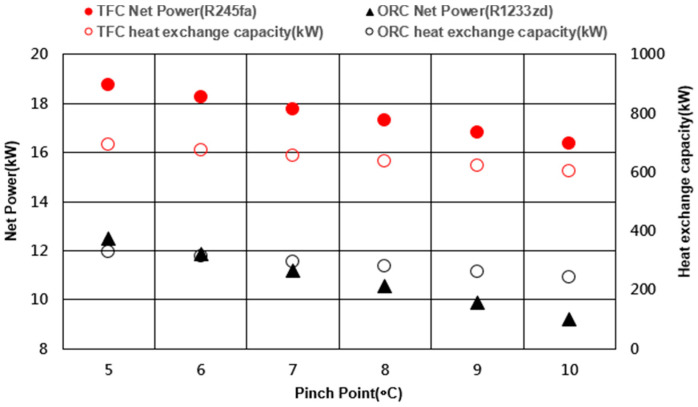
Performance of the TFC and ORC at different pinch points.

**Figure 12 entropy-21-01197-f012:**
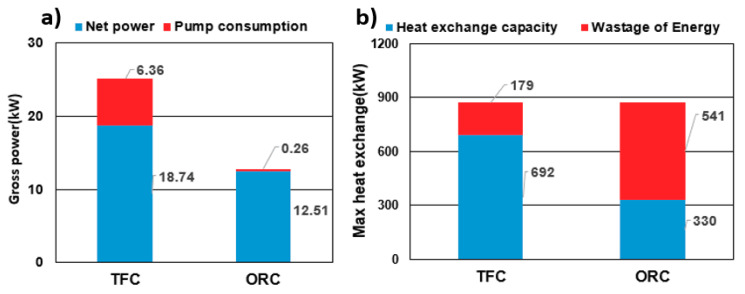
(**a**) The maximum net power generated by the TFC and ORC (ORC); (**b**) the maximum heat exchange by the TFC and ORC.

**Table 1 entropy-21-01197-t001:** Input data for the simulation case. ORC—organic Rankine cycle; TFC—trilateral flash cycle.

	Parameter	Value
Heat source	80 °C Water	4.16 kg/s
Heat sink	30 °C Water	26.6 kg/s
Cycle	Type	TFC, ORC
	Working Fluid	R245fa, R134a, R236fa, R1233zd
	Condensation temperature	37 °C
System Componet	Pump efficiency	0.7
	Expander efficiency	0.75
	Heat exchanger	Pinch temperature 5 °C
	Condenser	Pinch temperature 1 °C
	Pressure drop	No pressure drop

**Table 2 entropy-21-01197-t002:** Performance parameters and details for the most favorable TFC simulation.

Working fluid	R245fa	R134a	R236fa	R1233zd
Mass flow rate(kg/sec)	12.46	10.82	13.18	13.58
Evaporation(°C)	75	75	75	75
Evaporator pressure(kPa)	695	2364	1108	581
Heat exchange capacity(kW)	692	675	687	692
Gross power(kW)	25.08	33.95	28.37	24.42
Working fluid pump(kW)	6.35	18.88	10.08	6.03
Net power(kW)	18.73	15.06	18.28	18.38
Net Cycle Eff(%)	2.71	2.23	2.66	2.65
Power generation per unit mass	2.01	3.13	2.15	1.79

**Table 3 entropy-21-01197-t003:** Performance parameters and details for the most favorable ORC simulation.

Working fluid	R245fa	R134a	R236fa	R1233zd
Mass flow rate(kg/sec)	1.61	1.77	2.06	1.54
Evaporation(°C)	55	55	55	56
Evaporator pressure(kPa)	400	1491	668	349
Heat exchange capacity(kW)	348	348	347	330
Gross power(kW)	12.67	13.32	12.46	12.78
Working fluid pump(kW)	0.29	1.15	0.57	0.26
Net power(kW)	12.37	12.17	11.89	12.51
Net Cycle Eff(%)	3.55%	3.49%	3.42%	3.78%
Power generation per unit mass	7.86	7.52	6.04	8.29

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
