# Peer review of "Comparison of the Trilateral Flash Cycle and Rankine Cycle with Organic Fluid Using the Pinch Point Temperature"

_entropy, 2019, doi:10.3390/e21121197_

Round 1
Reviewer 1 Report
This paper presents a thermodynamic comparison between a trilateral flash cycle (TFC) and an organic Rankine cycle (ORC). The paper is well presented, the English language is easy to understand, but the methodology and the results are not interesting at all. There are thousands of publications that address this kind of simulations and they have more interesting results.
The reviewer found that the introduction is very light. The TFC is merely mentioned, the authors did not present a detailed literature review to establish the novelty and relevance of their work.
The weakness of the manuscript relies on the fixed parameters and the method that the authors are using: Simple energy balances based on fixed isentropic efficiencies (pump and turbine) which, by the way, are not supported.
To improve this work, the authors will require to change the method by including more straightforward thermodynamic and heat transfer equations to avoid the linear behaviour that makes the results predictable. In addition, the TFC has a two-phase expansion, which is a major concern in the design of the turbine and was not addressed in the manuscript.
Reviewer 2 Report
The cycles under comparison are Rankine Cycle and Triangular Flash Cycle. Both cycles are operated by Organic Working Fluid. Thus, the title must be changed because it gives wrong information (it seems that only the Rankine cycle is operated by organic fluid).
Figure 1 shows two cycles that give incorrect information because they have not been drawn taking one reference quantity into account. For a better understanding of the paper, what is the constant quantity has to be clarified and the drawing accordingly.
In Figure 1, the isentropic liquid compression temperature increase is in the same order of the real temperature increase. Due to the well-known physical behaviour of liquids, the isentropic temperature increase should be negligible. Moreover, the entropy increase of the heated organic fluid must be greater than the entropy decrease of the hot fluid. It is recommended to give in the abscissa the mass entropy change, not the specific entropy.
Equation 1 should be expressed in terms of pump pressure increase that influences the mass of the Rankine cycle organic working fluid as shown in Figure 2.
The efficiencies of the expanders of the two cycles have been assumed equal. In the Rankine cycle, the expansion takes place in the superheated vapour zone and in the triangular Flash cycle, the expansion occurs in the two-phase zone. This assumption is not correct. The TFC expander efficiency has to be discussed from the scientific point of view according to the liquid content at the end of the expansion.
Results have to be corrected accordingly.
Reviewer 3 Report
Technical appraisal of
Entropy 637091
Comparison of the trilateral flash cycle and organic Rankine cycle by using the pinch point temperature
This is a rather academic paper showing a potential way of getting some thermodynamic profit in the very corner of temperature. Practical feasibility of this proposal seems to be very low, or inexistent. Note the best theoretical efficiencies are in the range 3.5 through 4%, and machines actually working with such low values of steam title would be extremely ineffective.
So, if those effects are taken into account, and similar technical inefficiencies of pumping machines are also included in the final energy balance of the system, the final result can be negative, i.e., the system does not produce electricity: it consumes it.
If the axis of the TFC is directly used to the axis of a machine in a factory, the same can be found.
Authors must clarify the efficiency problem, and the availability of TFC turbines. They must also assess somehow the practical value of this idea. It is not only a problem of turbines. Pumps must properly be described, as well as evaporators.
In summary, the paper is very theoretical, dealing with the very corner of Thermodynamics. Authors, so to speak, are playing so close to the border, that they can be offside at any moment.
Round 2
Reviewer 1 Report
The authors significantly improved the manuscript. However, the literature review needs to be improved. It is not enough to say:
"but few researches present specific method for pinch point temperature"
By the way, the sentence above requires English editing.
The problem that you are addressing is "how to improve the thermal match between the temperature profiles of the heat source and the working fluid to reduce the irreversibilities in the heat exchanger due to finite temperature difference?". After explaining the previous work in this field, you have to introduce your topic, the pinch point. This is what you have to explain clearly.
I also suggest to include the following publications about TFC, ORC and the Goswami cycle and some publications from Entropi:
Lecompte, Steven, Martijn van den Broek, and Michel De Paepe. "Thermodynamic analysis of the partially evaporating trilateral cycle." 2nd International Seminar on ORC Power Systems (ASME Organic Rankine Cycle). 2013. Lecompte, Steven, et al. "Review of organic Rankine cycle (ORC) architectures for waste heat recovery." Renewable and sustainable energy reviews 47 (2015): 448-461. Fontalvo, Armando, et al. "Energy, exergy and economic evaluation comparison of small-scale single and dual pressure organic rankine cycles integrated with low-grade heat sources." Entropy 19.10 (2017): 476. Demirkaya, Gokmen, et al. "Experimental and theoretical analysis of the Goswami cycle operating at low temperature heat sources." Journal of Energy Resources Technology 140.7 (2018): 072005. Valencia, Guillermo, et al. "Energy and Exergy Analysis of Different Exhaust Waste Heat Recovery Systems for Natural Gas Engine Based on ORC." Energies 12.12 (2019): 2378. Fontalvo, Armando, et al. "Exergy analysis of a combined power and cooling cycle." Applied Thermal Engineering 60.1-2 (2013): 164-171. Demirkaya, Gokmen, et al. "Thermal and exergetic analysis of the goswami cycle integrated with mid-grade heat sources." Entropy 19.8 (2017): 416.
Author Response
Thanks for your comments, we have revised the manuscript and added a description of the research topic. In addition, we have also referred to the literature you provided and have cited several of them. We sincerely appreciate your suggestions.